# Amateur Female Athletes Perform the Running Split of a Triathlon Race at Higher Relative Intensity than the Male Athletes: A Cross-Sectional Study

**DOI:** 10.3390/healthcare11030418

**Published:** 2023-02-01

**Authors:** Guilherme Corrêa De Araújo Moury Fernandes, José G. G. Barbosa Junior, Aldo Seffrin, Lavínia Vivan, Claudio A. B. de Lira, Rodrigo L. Vancini, Katja Weiss, Beat Knechtle, Marilia S. Andrade

**Affiliations:** 1Sports Medicine Residency Program, Department of Orthopedics and Traumatology, Federal University of São Paulo, São Paulo 04021-001, São Paulo, Brazil; 2Department of Physiology, Federal University of São Paulo, São Paulo 04021-001, São Paulo, Brazil; 3Human and Exercise Physiology Division, Faculty of Physical Education and Dance, Federal University of Goiás, Goiânia 74690-900, Goiás, Brazil; 4Center for Physical Education and Sports, Federal University of Espírito Santo, Vitória 29075-210, Espirito Santo, Brazil; 5Institute of Primary Care, University of Zurich, 8091 Zurich, Switzerland; 6Medbase St. Gallen Am Vadianplatz, 9001 St. Gallen, Switzerland

**Keywords:** ventilatory threshold, $\dot{V}O_{2}max$, performance, respiratory compensation point, women

## Abstract

Maximal oxygen uptake ($\dot{V}O_{2}max$), ventilatory threshold (VT) and respiratory compensation point (RCP) can be used to monitor the training intensity and the race strategy, and the elucidation of the specificities existing between the sexes can be interesting for coaches and athletes. The aim of the study was to compare ventilatory threshold (VT), respiratory compensation point (RCP), and the percentage of the maximal aerobic speed (MAS) that can be maintained in a triathlon race between sexes. Forty-one triathletes (22 men and 19 women), 42.1 ± 8.4 (26 to 60) years old, that raced the same Olympic triathlon underwent a cardiorespiratory maximal treadmill test to assess their VT, RPC, and MAS, and race speed. The maximal oxygen uptake ($\dot{V}O_{2}max$) (54.0 ± 5.1 vs. 49.8 ± 7.7 mL/kg/min, *p* < 0.001) and MAS (17 ± 2 vs. 15 ± 2 km/h, *p* = 0.001) were significantly higher in male than in female athletes. Conversely, there were no sex differences according to the percentage of $\dot{V}O_{2}max$ reached at VT (74.4 ± 4.9 vs. 76.1 ± 5.4%, *p* = 0.298) and RCP (89.9 ± 3.6 vs. 90.6 ± 4.0%, *p* = 0.560). The mean speed during the race did not differ between sexes (12.1 ± 1.7 km/h and 11.7 ± 1.8 km/h, *p* = 0.506, respectively). Finally, men performed the running split at a lower percentage of speed at RCP than women (84.0 ± 8.7 vs. 91.2 ± 7.0%, respectively, *p* = 0.005). Therefore, male and female athletes accomplished the running split in an Olympic triathlon distance at distinct relative intensities, as female athletes run at a higher RCP percentage.

## 1. Introduction

Over the last few decades, there has been a notable increase in the number of female athletes [1], reaching its maximum at the 2020 Tokyo Olympics, with over 48% of the 11,300 athletes being women. Thus, the 2020 Tokyo Olympics were the most equal in terms of sex distribution. Specifically in the triathlon event, the participation of women has increased considerably during the last decades reaching between 25% and 40% of the total field [2].

In the triathlon, as in any other endurance sport, the main physiological determinants of performance are the maximum ability to absorb or utilize oxygen ($\dot{V}O_{2}max$), the fraction of $\dot{V}O_{2}max$ that might be sustained for long periods of time, and running economy [3]. Among these factors, $\dot{V}O_{2}max$ is certainly one of the most studied variables [4]. There is a relative consensus in the literature that the $\dot{V}O_{2}max$ is limited by the ability of the cardiorespiratory system to deliver oxygen to the exercising muscles, and not by the muscular capacity of extraction [3]. Previous studies have investigated sex differences in $\dot{V}O_{2}max$ between sedentary, amateur, and elite athletes, and all of them agree that female athletes present lower values than their male counterparts [5,6]. These differences are commonly attributed to different factors, such as higher body fat, lower red cell mass and hemoglobin levels, and lower end-diastolic, end-systolic, stroke volume, and cardiac output in women [7,8,9].

Other physiological variables that are commonly used as predictors of endurance performance are ventilatory thresholds [10]. During an incremental exercise test performed with the concomitant measurement of $\dot{V}O_{2}$, carbon dioxide production ($\dot{VC}O_{2}$), and minute ventilation, two distinct thresholds were identified [11]. The first, commonly called the ventilatory threshold (VT), is defined as the highest sustained intensity of exercise for which the measurement of oxygen uptake can account for the entire energy requirement. The second, called the respiratory compensation point (RCP), is characterized by the highest exercise intensity in which the body can buffer hydrogen ions production, preventing their accumulation [11,12]. There is relative consensus in the literature that the limiting factors for VT and RCP are peripheral conditions (extraction capacity) [5]. Considering that women present a greater proportional area of type I muscle fibers and preserve more glycogen than males, as they use more fatty acids, it is possible that female athletes present different VT and RCP than males. Although some recent studies have shown that the percentage of $\dot{V}O_{2}max$ at VT and RCP varies significantly between the sexes [13,14,15], the actual literature data are conflicting [5].

Considering that the speed of runs associated with $\dot{V}O_{2}max$, VT, and RCP can be used to monitor the training intensity and the race strategy [15], the elucidation of the specificities existing between the sexes can be interesting for coaches and athletes. Therefore, the aim of this study was to compare male and female athletes according to VT, RCP, and the percentage of the maximal aerobic speed that athletes can run in a triathlon race.

## 2. Materials and Methods

### 2.1. Participants and Study Design

Participant recruitment was carried out through direct contact with triathlon trainers and social media. Forty-one athletes participated in the study (22 male and 19 female). The male athletes were 41.4 ± 9.2 years old, 74.1 ± 6.9 kg and 174.2 ± 7.0 cm, and the female athletes were 42.8 ± 7.2 years old, 58.7 ± 6.6 kg and 163.9 ± 4.6 cm. The age was not significantly different between sex groups (*p* = 0.609) but male athletes were significantly heavier (*p* < 0.001) and taller (*p* < 0.001) than the female athletes. The inclusion criteria to take part in the study included having at least 3 years triathlon training experience, and being enrolled in the 32nd International Olympic Triathlon Santos, Brazil (Olympic distance). The exclusion criteria were not finishing the competition or having any medical condition that prevented the cardiorespiratory maximum treadmill test.

### 2.2. Experimental Procedures

All athletes underwent a cardiorespiratory incremental maximal test to determine $\dot{V}O_{2}max$, VT, and RCP, in addition to measure the maximal aerobic speed (MAS). All tests were conducted less than a month before the triathlon race competition between 20 January and 10 February 2022.

The cardiorespiratory incremental maximal test was performed using a treadmill (Inbrasport, ATL, Porto Alegre, Brazil) and computer-based metabolic analyzer (Quark, Rome, Italy). Before each test, the metabolic analyzer was calibrated according to the manufacturer’s guidelines. All the participants were instructed to avoid vigorous exercises the day before the test, and to avoid consuming stimulating beverages on the day of the test, such as coffee or tea. Additionally, the participants completed the Physical Activity Readiness Questionnaire (PAR-Q) [16]. Participants who did not answer “YES” to any of the PAR-Q questions and did not present any contraindication to participation were enrolled in the study. No-one answered “YES” to any question of the PAR-Q questions.

The test started with a 3 min warm up at 8 km/h, increased by 1 km/h in each minute until volitional exhaustion. The treadmill was programmed to have a 1% inclination to simulate the difficulties of open-air running [17]. Each test lasted between 8 and 12 min. Heart rate was monitored during the whole test using a heart rate monitor (Ambit 2S, Suunto, Finland), and perceived effort was rated according to the Borg scale (Noble et al., 1983).

The $\dot{V}O_{2}$, $\dot{VC}O_{2}$, O_2_ end-tidal pressure (PET O_2_), CO_2_ end-tidal pressure (PET CO_2_), and minute ventilation (${\dot{V}}_{E}$), were measured breath-by-breath, and all data were averaged over 20 s for analysis. VT and RCP were identified through the O_2_ and CO_2_ ventilatory equivalents and end-tidal pressures [18] by two independent investigators. In the case of discordance about the VT or RCP, a third investigator was consulted to identify these variables, so that, in all cases, there was agreement regarding the VT and RCP between at least two investigators. The speed at VT and RCP, in addition to the $\dot{V}O_{2}max$ percentage at VT and RCP have been presented. The $\dot{V}O_{2}max$ was determined as the stabilization of O_2_ consumption (increase lower than 2.1 mL · kg^−1^. min^−1^), even after increasing treadmill speed [19]. The maximal aerobic speed reached during the test was defined as the minimal speed eliciting $\dot{V}O_{2}max$ [20]. All the tests were collected by qualified professionals trained and experienced in the method.

### 2.3. Statistical Analysis

Data are presented as the mean and standard deviation. Descriptive analysis was conducted to evaluate the distribution of the variables. All variables presented normal distribution and homogeneous variability according to the Shapiro–Wilk and Levene tests, respectively. Student’s *t*-test was used to compare the mean values. The SPSS version 21.0 (SPSS, Inc., Chicago, IL, USA) was employed to perform the analysis. The G*Power version 3.1.9.2 (Franz, Universität Kiel, Germany) was used to determine the sample size and analyze the test power level. A sample size calculation on the velocity maintained during the running split, using data from a pilot study (n = 12), which are 11.0 ± 1.0 km/h for male athletes and 10.2 ± 0.9 km/h for female athletes, showed that 19 athletes in each group (male or female) were needed to detect a relevant difference with 80% power and a significance level of 5%. For power level calculation, a *t*-test family was selected, and mean values, standard deviations, and effect sizes (Cohen *d*) were included in the calculation. The power of the test varies from 0 to 1. Usually, researchers use 0.80 as the power level of the test [21]. The measurement of the effect size for differences between sexes were determined by calculating the mean difference between the two sexes, and then dividing the result by the pooled standard deviation. Calculating effect sizes, the magnitude of any change was judged according to the following criteria: *d* < 0.2 considered no effect, 0.2 ≤ *d* < 0.5 considered a “small” effect size; 0.5 ≤ *d* < 0.8 represented a “medium” effect size; and *d* ≥ 0.8 a “large” effect size [22]. The level of significance was set at *p* < 0.05.

## 3. Results


The absolute and relative to body mass values for $\dot{V}O_{2}max$ and the maximal aerobic speed were significantly higher for male than for female athletes. Conversely, there were no sex differences in the percentage of $\dot{V}O_{2}max$ reached at VT and RCP. However, the speeds at VT and RCP were higher in male athletes (Table 1).Despite the sex differences according to $\dot{V}O_{2}max$, VT, and RCP, the mean speed maintained during the running split of an Olympic triathlon race was not different between sexes (*p* = 0.506, *d* = 0.23). The running speed during the race was situated between the speeds associated with VT and RCP for both sexes. However, female athletes performed the running split at a higher percentage of the speed at the RCP than male athletes (Table 1).


## 4. Discussion

The main findings of the present study were the following: (i) female and male amateur athletes presented similar VT and RCP related to $\dot{V}O_{2}max$; (ii) male athletes presented higher speed at VT and RCP than female athletes; (iii) the mean running speed maintained during the running split of a triathlon race did not differ between sexes; and (iv) female athletes maintained the running split at a higher relative intensity considering the percentage of RCP, than male athletes.

The running split represents approximately 25% of the overall race time of a triathlon race, although it has been demonstrated to be the main determinant in Olympic distance triathlon overall performance [23,24] for male and female athletes. Considering the importance of the running split for triathlon performance, several previous studies have investigated the determinant factors for this split time [25,26]. However, little is known about sex-related differences.

According to $\dot{V}O_{2}max$, which has been defined as the highest rate at which oxygen can be taken up and utilized by the body during severe exercise [3], the results of the present study corroborate previous findings that female athletes present lower $\dot{V}O_{2}\;max$ than male athletes [27]. In the same direction, Puccinelli et al. [13] also showed higher $\dot{V}O_{2}max$ values for male than for female triathletes (59.9 ± 6.3 and 49.5 ± 7.8 mL/kg/min, respectively). The $\dot{V}O_{2}max$ is limited by central cardiovascular factors [3]; therefore, the smaller cardiac volume, cardiac output, and the lower hematocrit level, are possibly associated factors to the lower $\dot{V}O_{2}max\;$ for female athletes [28].

Furthermore, the present results showed no sex differences in %$\dot{V}O_{2}max$ at VT and RCP. The data on sex differences in VT and RCP are conflicting [5,13]; VT and RCP are limited by peripheral conditions (e.g., muscle adaptations to aerobic exercises [3,29], such as increased capillary density, the increased mitochondrial content of muscle, and higher mitochondrial enzyme levels (i.e., succinate dehydrogenase, NADH dehydrogenase, and NADH-cytochrome c reductase)). This limitation can be explained because to have the necessary energy to muscular contraction, ATP is converted to ADP and Pi, and the high levels of ADP and Pi drive metabolic reactions to resynthesize new ATP molecules to continue the muscular contraction [3]. With few mitochondria in a muscle cell, ADP levels should increase substantially to reach the ADP demand via aerobic pathways, and high ADP levels also have a stimulatory effect on phosphofructose kinase (PFK), stimulating glycolysis via and generating a high demand for carbohydrates as an energetic substrate [3]. In a muscle with higher mitochondrial content, the ADP level increases less for the same $\dot{V}O_{2}$ demand; therefore, there is less PFK stimulus and less carbohydrate turnover, increasing the possibility of using fat as an energetic substrate, resulting in later lactate formation [3,29].

Considering that women show a greater proportional area of type I muscle fibers [5], higher VT and RCP could be expected in female athletes. Indeed, Puccinelli et al. [13] demonstrated that female amateur athletes presented with higher VT than male athletes. In addition, Puccinelli et al. [13] selected only amateur triathletes; only 18 women and 39 men were included in the study. Considering sex differences in the willingness to participate in certain types of research [5], we cannot exclude the hypothesis that only the best-trained female athletes volunteered to participate in the study; therefore, the different levels of training could be a reason for the different %$\dot{V}O_{2}max\;$ at VT and RCP, and not the sex metabolic differences. In a recent review, Besson et al. [5] reported that there were no sex differences according to %$\dot{V}O_{2}max$ at ventilatory thresholds. As expected, despite the lack of sex differences according to the %$\dot{V}O_{2}max$ at ventilatory thresholds, as the male athletes presented significantly higher $\dot{V}O_{2}max$ and MAS, the speed at the ventilatory thresholds was higher for men than for women.

Furthermore, it is also generally accepted that carbohydrate feeding during prolonged exercises (longer than 2 h) can maintain the possibility of carbohydrate oxidation and the prevention of hypoglycemia [30]. Beyond that, a good level of hydration during prolonged exercise is of fundamental importance, once it has been demonstrated that dehydration steadily increased body temperature, heart rate, and oxygen uptake for the same speed of running, worsening the physical performance [31].

The percentage of running speed at RCP that was sustained during the running split of the triathlon race was significantly higher in the female group. This was an interesting finding. As women were able to sustain a higher percentage speed at RCP, the average speed maintained during the race did not differ between the sexes, and consequently the race time. There are some possible explanations for why a woman can maintain a higher percentage of RCP speed than a man during a 10 km run. As stated previously, female athletes showed a greater proportional area of type I fibers. In addition, women can spare more carbohydrates during exercise, as they are more capable in using fatty acids as an energetic substrate during aerobic exercises, as documented by Tarnopolsky et al. [32], who found lower respiratory exchange ratio values were found for women (0.87 × 0.94). Moreover, despite the lack of evidence, women appear to have some advantage in the biomechanics of running, especially due to shorter stride length and higher stride frequency, and consequently, a lower duty factor [33]. Women have also been shown to have a different neuromotor strategy, which could reduce their fatigability during endurance exercises. It was suggested that experienced ultra-endurance-trail women were more resistant to fatigue after ultra-trail running, as they had a lower decrease in maximal voluntary torque changes in the knee extensors and plantar flexors [34]. These differences make women more resistant to fatigue during long-term aerobic exercises [5].

In the present study, the swimming and cycling intensities of male and female athletes were not measured. It is possible that the athletes performed the first division (swimming and cycling) at different relative intensities, which could have affected the results of the running split. However, this does not invalidate the result that men and women perform running splits at different relative intensities. In the present study, the level of hydration of the athletes and carbohydrate feeding were not evaluated. Considering that a possible dehydration or hypoglycemia of the athletes can affect sports performance, this can be considered a limitation of the present study. The authors suggest that future studies should be developed to assess the relative intensity in which men and women perform the three triathlon splits.

## 5. Conclusions

Despite having no difference according to VT and RCP, male and female athletes performed the running split in an Olympic triathlon distance at different relative effort intensities, and female athletes ran at a higher relative intensity (i.e., a higher RCP percentage). These findings can be useful for coaches and athletes to consider sex differences when designing the strategy for a triathlon race, which may be different for each sex.

## Figures and Tables

**Table 1 healthcare-11-00418-t001:** Comparison between sexes of the descriptive characteristics of athletes.

	Male (n = 22)	Female (n = 19)	*p*-Value	Power (1-ß)	Effect Size (*d*)	CI for Effect Size
$\dot{V}O_{2}max$ (L/min)	4.00 ± 0.52	2.89 ± 0.40	<0.001	1.00	2.39	1.3 to 3.4
$\dot{V}O_{2}max$ (mL/kg/min)	54.0 ± 5.1	49.8 ± 7.7	0.047	0.64	0.64	0.1 to 1.2
MAS (km/h)	17 ± 2	15 ± 2	0.001	0.93	1.00	0.6 to 1.4
$\%\dot{V}O_{2}\;max$ at VT	74.4 ± 4.9	76.1 ± 5.4	0.298	0.26	0.32	−0.2 to 0.8
Speed at VT (km/h)	11.8 ± 1.1	10.7 ± 1.5	0.021	0.83	0.83	0.3 to 1.4
$\%\dot{V}O_{2}\;max$ at RCP	89.9 ± 3.6	90.6 ± 4.0	0.560	0.14	0.18	−0.3 to 0.7
Speed at RCP (km/h)	14.3 ± 1.2	12.8 ± 1.6	0.001	0.95	1.06	0.6 to 1.5
Mean speed in running split (km/h)	12.1 ± 1.7	11.7 ± 1.8	0.506	0.18	0.23	−0.4 to 0.8
% RCP maintained during the running split	84.0 ± 8.7	91.2 ± 7.0	0.005	0.88	0.91	0.4 to 1.4

Data are presented as mean ± standard deviation. $\dot{V}O_{2}max$: maximal oxygen uptake; MAS: maximal aerobic speed; VT: ventilatory threshold; RCP: respiratory compensation point; CI: confidence interval.

## Data Availability

Data supporting the study results can be provided followed by a request sent to the corresponding author’s e-mail.

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
