# Peer review of "Amateur Female Athletes Perform the Running Split of a Triathlon Race at Higher Relative Intensity than the Male Athletes: A Cross-Sectional Study"

_healthcare, 2023, doi:10.3390/healthcare11030418_

Round 1

Reviewer 1 Report

It may be but (not compulsory) included more reference info regarding ?̇?2???, VT, and RCP from the monitoring perspective.

Author Response

It may be but (not compulsory) included more reference info regarding ?̇?2???, VT, and RCP from the monitoring perspective.

Answer: Thank you for your suggestion. A Systematic review with meta-analysis about the interval training has been included. Please let us know if this modification does not resolve your doubts in this matter.

Reviewer 2 Report

Title: A bit long but include all key elements. In my humble view, it does not need changes.

Abstract: Include a first sentence that justifies this study. Review “maintain” instead of “maintained”. Indicate the range of sample ages.

Keywords: Do not repeat them from those which already are in title, so as to boost the visibility of this paper in the different databases this journal is indexed in.

Introduction: The first four quotes are too old to try to justify something that is recent. Update them (use studies from 2020 to 2023 when possible). Include a quote to justify the sentence in lines 66-68.

Method: The access to the sample must be deeply extended. More details are required. Their ages, their sex, their origins… Was this study design approved by any ethical committee? In line 88, which specific triathlon race are you referring to? Review the quote in line 100. What training or experience in collecting this type of data did investigators (line 104-105) have?

Results: if possible, avoid splitting tables into two different pages. Delete 0 in those values that can vary from 0.001 to 1.

Discussion: Do not limit discussion section to indicate if these results confirm or not previous one but try to explain them based on scientific literature and to develop its implications to the field of knowledge. Review the quote in lines 156, 178, 180, 198. Consider more limitations and future research lines as well.

Author Response

Title: A bit long but include all key elements. In my humble view, it does not need changes.

Answer: Thank you for your positive comment.

Abstract: Include a first sentence that justifies this study. Review “maintain” instead of “maintained”. Indicate the range of sample ages.

Answer: Thank you about your suggestions. A sentence that justifies the study has been included, the word “maintained” has been corrected and the range of sample ages has been added. Please let us know if these modifications do not resolve your doubts in this matter.

Keywords: Do not repeat them from those which already are in title, so as to boost the visibility of this paper in the different databases this journal is indexed in.

Answer: Thank you for your insightful comment. The keywords have been changed as requested by you.

Introduction: The first four quotes are too old to try to justify something that is recent. Update them (use studies from 2020 to 2023 when possible). Include a quote to justify the sentence in lines 66-68.

Answer: We totally agree with the reviewer. More recent references have been included and also a quote was included in lines 66-68.

Method: The access to the sample must be deeply extended. More details are required. Their ages, their sex, their origins… Was this study design approved by any ethical committee? In line 88, which specific triathlon race are you referring to? Review the quote in line 100. What training or experience in collecting this type of data did investigators (line 104-105) have?

Answer: Thank you about your constructive comments.

 The age, weight, height and sample size for both sexes have been included.

The study design was approved by the Human Research Ethics Committee of the Federal University of São Paulo and conformed to the principles outlined in the Declaration of Helsinki (approval number 0973/2021). This information was included in the end of the manuscript, after the conclusion section.

In line 88, we are referring to an Olympic distance triathlon race and this information has been included in the manuscript.

The quote in line 100 has been reviewed.

All the tests were collected by qualified professionals trained and experienced in the method. This information has been included in the manuscript.

Please let us know if these modifications do not resolve your doubts in this matter.

Results: if possible, avoid splitting tables into two different pages. Delete 0 in those values that can vary from 0.001 to 1.

Answer: The table has been readjusted. Thank you.

Discussion: Do not limit discussion section to indicate if these results confirm or not previous one but try to explain them based on scientific literature and to develop its implications to the field of knowledge. Review the quote in lines 156, 178, 180, 198. Consider more limitations and future research lines as well.

Answer: Thank you about your constructive comment and insightful suggestions. At the end of the third paragraph there are possible reasons for the difference in VO2max between genders. The limiting factors for ventilatory thresholds and their possible effects on the sex differences are also discussed in the fourth and fifth paragraphs. The sixth paragraph was included to discuss other factors that may have influenced the results, such as hydration level and carbohydrate consumption. Other limitations of the study are included in the last paragraph of the discussion. The study results showed that the male and female athletes can perform a triathlon race at different relative running intensity, therefore the training prescription and race strategy could be different between sexes. This information was included in conclusion section.  All the quotes and references list were reviewed, thank you for drawing our attention to this.

Please let us know if this modification does not resolve your doubts in this matter.

Reviewer 3 Report

See attached

Author Response

Reviewer #3

Page 2 line 47 – The sentence has been rewritten as requested by you.

Page 2 line 55 - The sentence has been rewritten as requested by you. Women presented lower end-diastolic, end-systolic and stroke volume than the male. This information has been included in the text in order to clarify.

Page 2, line 62 - The sentence has been rewritten as requested by you. The ventilatory threshold (VT) is defined as the highest sustained intensity of exercise for which measurement of oxygen uptake can account for the entire energy requirement.

Page 2, line 70 – The phase has been rewritten to clarify and meet with your expectation.

Page 2 line 74 – Yes, we agree with the expert review. It is a cross sectional design, which is clarified in the title of the manuscript and methods section.

Page 2 line 75 – In the methods section we clarify that the mean speed over the running split has been used to the analysis.

Page 2, line 81 - The inclusion criteria has been rewritten. The inclusion criteria to take part to the study included having at least three years triathlon training experience and being enrolled in the Brazilian Triathlon Trophy (Olympic distance).

Page 2, line 87 – Yes, we agree with the reviewer that the MAS is the speed that VO2 max was established at. However, it is an important variable to be analyzed, therefore we chose to maintain it.

Page 4 line 105 – The sentence has been rewritten. The speed at VT and RCP, in addition to the VO2max percentage at VT and RCP have been presented.

Table 1. Yes, we agree with the expert reviewer. It seems that the female athletes buffered ions better during the race than the male athletes.

Page 5 line 160 – The sentence has been rewritten.

Page 6 line 170 – Consideration about the hydration or carbohydrate intake are included in the manuscript.

Page 7 line 202 – The word “running” has been included.

Page 7 line 200 - In fact, water and carbohydrate replacement were not controlled during the race. Considering that these factors can affect the intensity of effort that can be maintained during the race, these factors were added to the limitations of the study.

Page 7 line 207 – The physical training level of the volunteers has been included in the sentence.

Please let us know if these modifications do not resolve your doubts.

Reviewer 4 Report

The authors researched an interesting field of exercise physiology which is important to understand energy changes, fatigue and athletic performance in endurance sport. However, we recommend that the authors should consider the following suggestions to improve their paper.

Methods

Study design

-The authors need to clearly explain sample selection and size. 

-There is no age hence, this is difficult to make the reader consider the sample as valid. 

-While Informed consent was sort, authors should include that participants were free to opt any stage of the experiment without giving reasons for their decision.

Experimental procedures

- Authors are encourage to discuss the pre-experimental procedure. 

- Steps of the experiment should be clearer that is it usable by any other researcher intending to use the same procedure in a different context. 

-The authors need to explain what they imply., on page 3, lines 104-5, ''In case of discordance, a third investigator was consulted''.

Statistical Analysis

-This paragraph needs to be re-arranged so that it can flow well. 

- We suggest that the authors use 'gender'  to 'sex'

Results

-Considering the biological differences, male athletes have high V02max compared to their female counterparts, hence these results are too obvious as the authors mentioned in lines 127 -9 page 3.  

Discussion

Intext referencing should be uniform. Check page 4 line 156, 180, 185, 198. 

References

Authors are advised to follow the journal referencing style which includes DOI. 

Author Response

Reviewer #4

Comments and Suggestions for Authors

The authors researched an interesting field of exercise physiology which is important to understand energy changes, fatigue and athletic performance in endurance sport. However, we recommend that the authors should consider the following suggestions to improve their paper.

Answer: Thank you about your positive and constructive comments.  

Methods

Study design

-The authors need to clearly explain sample selection and size. 

Answer: Participant recruitment was carried out through direct contact with triathlon trainers and social media. A sample size calculation on the speed maintained during the running split showed that 19 athletes in each group (male or female) were needed to detect a relevant difference with 80% power and a significance level of 5%. This information was included in the main manuscript in order to clarify and meet with your expectation.

-There is no age hence, this is difficult to make the reader consider the sample as valid. 

Answer: Thank you for drawing our attention to this point. The male athletes were 41.4±9.2 years old, 74.1±6.9 kg, and 174.2±7.0 cm; and the female athletes were 42.8±7.2 years old, 58.7±6.6 kg and 163.9±4.6 cm. This information has been included in the main manuscript in order to clarify.

-While Informed consent was sort, authors should include that participants were free to opt any stage of the experiment without giving reasons for their decision.

Answer: This information was included in the manuscript (Institutional Review Board Statement) as requested by you.

Experimental procedures

- Authors are encourage to discuss the pre-experimental procedure. 

Answer: We totally agree with the reviewer. All the participants were instructed to avoid vigorous exercises the day before the test to avoid consuming stimulating beverages on the day of the test, such as coffee or tea. Additionally, prior to the test the participants completed the Physical Activity Readiness Questionnaire (PAR-Q). Participants who did not answer “YES” to any of the PAR-Q questions and did not present any contraindication to participation were enrolled in the study. No one answered “YES” to any question of the PAR-Q questions. The information has been included in the experimental procedure section in order to clarify and meet with your expectation.

- Steps of the experiment should be clearer that is it usable by any other researcher intending to use the same procedure in a different context. 

Answer: A cardiorespiratory incremental maximal test started with a 3-minute warm up at 8 km/h, increased by 1 km/h in each minute until volitional exhaustion. The treadmill was programmed to have a 1% inclination to simulate the difficulties of open-air running. Each test lasted between 8 and 12 minutes. Heart rate was monitored during the whole test using a heart rate monitor (Ambit 2S, Suunto, Finland), and perceived effort was rated according to the Borg scale. All this information was included in the experimental procedure section in order to clarify and meet with your expectation.

-The authors need to explain what they imply., on page 3, lines 104-5, ''In case of discordance, a third investigator was consulted''.

Answer: Thank you about your constructive comment. In case of discordance about the VT or RCP, a third investigator was consulted to identify these variables, so that, in all cases, there was agreement regarding the VT and RCP between at least two investigators.

Statistical Analysis

-This paragraph needs to be re-arranged so that it can flow well. 

Answer: The paragraph has been rewritten and other information has been included.

- We suggest that the authors use 'gender'  to 'sex'

Answer: Thank you for your comment. However, we respectfully disagree with you. Sex is usually categorized as female or male but there is variation in the biological attributes that comprise sex and how those attributes are expressed, and gender refers to the socially constructed roles, behaviors, expressions and identities of girls, women, boys, men, and gender diverse people (Short et al 2013, Dotto 2019). For these reasons, we chose to keep the term sex, since this was what was asked of the participants.

Dotto GP. Gender and sex—time to bridge the gap. EMBO Mol Med. 11(5), e10668, 2019.

Short S.E., Yang Y.C., Jenkins T.M. Sex, Gender, Genetics, and Health. Am J Public Health.103 (Suppl 1), S93-101, 2013.

Results

-Considering the biological differences, male athletes have high V02max compared to their female counterparts, hence these results are too obvious as the authors mentioned in lines 127 -9 page 3.  

Answer: We agree with the expert reviewer, and that are also several previous studies showing higher VO2max values for male than for female athletes. However, the difference in VO2max between the sexes, although obvious, is an important positive control of the study and shows that the recruited participants had expected characteristics due to the results traditionally presented in the literature. Moreover, we also chose to maintain this result because we want to present the effect size of the sex on VO2max values, VAM, VT and RPC, with are difference variables associated with aerobic performance. We would like that the reader to be able to get a broader view of the sex differences of all performance-associated variables that were measured in the study.

Discussion

Intext referencing should be uniform. Check page 4 line 156, 180, 185, 198. 

Answer: Thank you for drawing our attention to this point. All the references were checked. Sorry about the mistake.

References

Authors are advised to follow the journal referencing style which includes DOI. 

Answer: Thank you for drawing our attention to this point. All the references were checked and the DOI numbers were included.

Please let us know if these modifications do not resolve your doubts.

Round 2

Reviewer 2 Report

Dear authors,

in my humble view, this version is much better than the previous one. I only have two more comments. Review the quote in line 148. In lines 184-185 "Clique ou toque aqui para inserir o texto" is written.

Author Response

Reviewer #2

Dear  authors,

In my humble view, this version is much better than the previous one. I only have two more comments. Review the quote in line 148. In lines 184-185 "Clique ou toque aqui para inserir o texto" is written.

Answer: Thank you for drawing our attention to this point. In fact, quote in line 148 was wrong. The sample size was calculated from data obtained from a pilot study. The sentence has been corrected and the values used to calculate the sample size have been included in the text. Additionally, the phrase "Click or tap here to enter text" has been deleted. Please let us know if this modification does not resolve your doubts in this matter.

Reviewer 3 Report

No further comment

Author Response

Reviewer #3

No further comment

Answer: no changes are required

Reviewer 4 Report

All comments are in the attached document. 

Author Response

Reviewer #4

All comments are in the attached document. 

Answer: All the comments and references have been corrected in the main text. Please let us know if the modifications do not resolve your doubts.